# Detection of Farmland Obstacles Based on an Improved YOLOv5s Algorithm by Using CIoU and Anchor Box Scale Clustering

**DOI:** 10.3390/s22051790

**Published:** 2022-02-24

**Authors:** Jinlin Xue, Feng Cheng, Yuqing Li, Yue Song, Tingting Mao

**Affiliations:** College of Engineering, Nanjing Agricultural University, Nanjing 210031, China; 2020112010@stu.njau.edu.cn (F.C.); 2018112011@stu.njau.edu.cn (Y.L.); 2020212008@stu.njau.edu.cn (Y.S.); 2019112010@stu.njau.edu.cn (T.M.)

**Keywords:** farmland obstacles, target detection, YOLOv5s, deep learning

## Abstract

It is necessary to detect multi-type farmland obstacles in real time and accurately for unmanned agricultural vehicles. An improved YOLOv5s algorithm based on the K-Means clustering algorithm and CIoU Loss function was proposed to improve detection precision and speed up real-time detection. The K-Means clustering algorithm was used in order to generate anchor box scales to accelerate the convergence speed of model training. The CIoU Loss function, combining the three geometric measures of overlap area, center distance and aspect ratio, was adopted to reduce the occurrence of missed and false detection and improve detection precision. The experimental results showed that the inference time of a single image was reduced by 75% with the improved YOLOv5s algorithm; compared with that of the Faster R-CNN algorithm, real-time performance was effectively improved. Furthermore, the *mAP* value of the improved algorithm was increased by 5.80% compared with that of the original YOLOv5s, which indicates that using the CIoU Loss function had an obvious effect on reducing the missed detection and false detection of the original YOLOv5s. Moreover, the detection of small target obstacles of the improved algorithm was better than that of the Faster R-CNN.

## 1. Introduction

The development of intelligent agricultural machinery and unmanned agricultural machinery has put forward higher requirements for real-time and accurate farmland obstacle detection. Because the farmland environment is open, there are often multiple obstacles, and single obstacle detection cannot effectively detect all obstacles, which will bring security risks to farmland operations. At present, multi-obstacle detection technology is mainly used in the field of automatic driving of automobiles, and it is used less in agriculture [1,2]. In order to ensure the safe driving and operation of unmanned agricultural machines, multi-obstacle detection technology is essential.

At present, the detection of obstacles in farmland mainly relies on two tools: radar and visual sensors. Considering the cost and the complex environmental factors of farmland, vision sensors are more suitable for obstacle detection in farmland than radar. Vision sensors can provide rich obstacle information, including category information, shape information, color information, and texture information of the target obstacle [3,4]. Most obstacle detection technologies based on visual sensors focus on indoor environments or simple outdoor environments with known structure (such as structured roads, expressways and urban streets with better conditions), and the research on farmland environments is not more extensive [5]. The literature shows that traditional methods mostly use stereovision technology, Histogram of Oriented Gradients (HOG) features and support vector machines (SVM) to detect obstacles in farmland [6,7,8,9]. The disadvantages of traditional methods are as follows:(1)Obstacles are only detected and located, but obstacles cannot be identified and classified, which is disadvantageous to the accurate path planning and obstacle avoidance of agricultural robots or unmanned agricultural vehicles.(2)The types and number of detected obstacles are limited, and if the selected features are not enough to represent the target obstacle, the missed or fail detection rate will be increased.

The rise of deep learning has brought new ideas for the detection and identification of obstacles in farmland [10,11,12,13,14]. For obstacle detection in the farmland environment, the Faster R-CNN detection algorithm and the YOLOv5s detection algorithm with the fastest detecting speed are used, respectively. Since the depth of Faster R-CNN is greater than that of YOLOv5s, the detection accuracy of Faster R-CNN is higher than that of YOLOv5s. However, with the increase in depth, the size of the weight file will also increase, and the inference speed will be slower, which is not conducive to the deployment of smart or unmanned agricultural machines. The current YOLOv5s detection speed is fast, but accuracy is relatively low.

Unmanned agricultural machinery may encounter various farmland obstacles such as people, agricultural vehicles, wire poles, farmhouses, trees, livestock and so on, which requires it to detect obstacles with faster speed and higher accuracy. Therefore, the motivation of this study is to propose a more suitable algorithm for the multi-type farmland obstacle detection of unmanned agricultural machinery. Furthermore, the innovation of the algorithm is suggested to improve the original YOLOv5s algorithm in two ways: one is to use the K-Means algorithm to speed up the convergence speed of training and inference by improving the initial anchor frame, and another is to use the CIoU Loss function that combines the three geometric measures of overlapping area, center point distance and aspect ratio to reduce the occurrence of missed or false detection and improve detection accuracy. The improved algorithm is conducive to the real time functionality and accuracy requirements of unmanned agricultural machinery for farmland obstacle detection. The rest of the structure of this paper is as follows: Section 2 shows the related work of YOLOv5s obstacle detection research. Section 3 introduces the materials and methods of this study, including the production of farmland obstacle data sets, the improvement of YOLOv5s and its training and reasoning. Section 4 shows the experimental results of this study, mainly including the comparison between the improved YOLOv5s and the original YOLOv5s and Faster R-CNN. This section also introduces the comparison results of the improved YOLOv5s model in this study with improved models in the related literature. Finally, conclusions are drawn in Section 5.

## 2. Related Work

In recent years, research on the YOLOv5 algorithm has involved many fields, such as fruit-picking identification in agriculture, cancer detection and identification in medicine, walking assistance for visually impaired people, and the identification of surface or underwater targets.

Researchers in agriculture currently focus on the picking of agricultural products. In the literature [15], research has improved the bottleneck CSP(Cross Stage Partial) module and added the SE(Squeeze-and-Excitation) module in YOLOv5s to achieve a lightweight model, so as to achieve the purpose of fast and accurate identification of apples. In the literature [16], the improved CBF (Counting Bloom Filter) module based on CBF and a Specter module replacing CSP module were proposed to realize the lightness of YOLOv5s model, and improve the detection speed and accuracy of picking Zanthoxylum fruit. In terms of medicine [17], the LSCC (Laryngeal Squamous Cell Carcinoma) using the integrated algorithm (YOLOv5s + Yolov5M-TTA) had the best results, and its performance indicators were comparable to those reported by other state-of-the-art detection models. In the literature [18], an assist system including embedded system jetson AGX Xavier and binocular depth camera zed 2 was proposed to help visually impaired people walk outdoors. The system identified obstacles in front of users through the YOLOv5s network, effectively guiding visually impaired people to walk safely on sidewalks and crosswalks. Aiming at the problem of real-time monitoring of floating waste in the waterway [19], a FMA (feature map attention) layer was added to improve YOLOv5s, and splicing data enhancement technology was used to improve the training of small and medium-sized target detection effects. In the literature [20], the TR-YOLOv5s network and down sampling principle were proposed, and an attention mechanism was introduced to meet the high-precision and high-efficiency identification requirements for underwater targets. The literature [21] added the Ghost bottleneck lightweight depth convolution module to the Backbone and Neck modules of YOLOv5 so the model size was reduced, and at the same time, the SELayer module of the attention mechanism was added to the Backbone module, and the deep convolution (DWConv) was used to compress the network parameters to achieve fast and accurate identification of particleboard defects.

The above literature, whether in the agricultural field or non-agricultural field, focuses basically on single-type target detection and recognition, while this paper mainly considers the detection and recognition of multi-type farmland obstacles. Considering the limited ability of the calculation of on-board equipment of unmanned agricultural machinery and the complexity of the farmland environment, the proposed algorithm is improved in two aspects: using K-Means clustering to speed up convergence and thus to improve single image detection speed; and using CIoU Loss to reduce missed detection and false detection. Table 1 shows a comparison between the YOLOv5s algorithms used in agriculture and the proposed algorithm in this paper.

## 3. Materials and Methods

### 3.1. Dataset Creation

This study focuses on images of 8 kinds of common farmland obstacles in farmland, including people, tractors, haystacks, wire poles, trees, houses, sheep and cattle. These images were collected from the photography collection of farmlands and the Internet. In this way, a dataset consisting of 2000 photos taken from farmland by using a mobile phone (HUAWEI Co. Ltd., Shen Zhen, China) and 370 pictures collected from the Internet, totaling 2370 images, was formed. Then, these images were used as training sets and test sets with a ratio of 8:2.

The labeling software LabelImg was used to label 8 kinds of common farmland obstacles in the images, including people, tractors, haystack, wire poles, trees, houses, sheep and cattle. The labeling process is shown in Figure 1. All obstacles were taken as background except the labeled targets, and the labeling format was of the standard PASCAL VOC2007 format. The total number of labeling boxes containing various obstacles in 2370 images was up to 5405, and the number of labeling boxes containing each type of obstacle is shown in Figure 2.

When training with the dataset of farmland obstacles with small samples, the deep learning model is prone to overfitting, so it is not enough to only use the collected dataset. However, the acquisition of farmland obstacles is time-consuming and labor-intensive, while currently there is a lack of large-scale databases related to farmland obstacles. Therefore, expanding the dataset by data enhancement is a common technique to alleviate the data shortage problem. It expands the training dataset through image transformation, such as flipping, rotation, scaling, and color adjustment [22].

In the farmland environment, changes in light and extreme weather such as rainy days or foggy days have a certain impact on the image acquisition of the vision sensor. In this paper, data enhancement of random brightness was used to simulate changes in light, and Gaussian noise, rainy processing and atomization processing were used to simulate rainy or foggy days. According to the different shooting angles, the images were simulated by random angle rotation. Therefore, the data of the training set was extended to 14,220, and the expansion processing of an image is shown in Figure 3.

### 3.2. Original YOLOv5s Network Architecture

The YOLOv5s network structure is shown in Figure 4, including four processing stages: Input, Backbone, Neck and Prediction [23]. Compared with YOLOv3 and YOLOv4, YOLOv5 has Mosaic data enhancement and adaptive anchor box calculation functions at the input, Focus structure and CSP structure in Backbone [24], PANnet in Neck and GIoU Loss in Prediction. The improvements in this paper include using the K-Means clustering algorithm in Input and replacing GIoU Loss with CIoU Loss in Prediction.

### 3.3. Improved YOLOv5s Network Architecture

The initial anchor box scale of the original YOLOv5s is based on the COCO dataset. When the anchor box scale is applied to the farmland obstacle dataset, it cannot cover all target obstacle scales. As a result, the parameter learning time is prolonged and the detection speed is reduced. Meanwhile, the frequency of missed detection and false detection is high for the original YOLOv5s with *GIoU* Loss function. Therefore, an improved YOLOv5s algorithm was proposed. First, the K-Means clustering algorithm was used to determine the initial scale of the anchor box covering all scales as much as possible. Second, *CIoU* Loss function instead of *GIoU* Loss function was used to calculate the loss to reduce the missed detection or false detection rate.

#### 3.3.1. K-Means Clustering Algorithm

Generally, each network generates anchor boxes with different fixed scales to detect obstacles with different scales. Initial anchor boxes with different scales could cover part of the scales of target obstacles, but for target obstacles in the acquired images from farmland whose aspect ratios are inconsistent with the images from the COCO dataset and small-scale target obstacles, the network needs to learn further parameters, which leads to slow convergence speed during model training. Therefore, the K-Means clustering algorithm for generating anchor box scales was integrated into the YOLOv5s detection framework in this study.

The K-Means clustering algorithm divides the dataset into several different clusters according to certain rules, and the similarity of the same cluster is high while that of different clusters is low [25]. The workflow of the K-Means clustering algorithm is shown in Figure 5.

Firstly, *k* anchor boxes were randomly selected from different initial anchor box scales as cluster centers, and the sample mean of the clusters was calculated to find its own center point for each sample and update the cluster centers of the anchor box scales. Then, it was judged whether clustering before and after was the same for the samples. If it was different, clustering would be continued. Otherwise, the initial anchor box scale after clustering was output [26].

Since the size of each image in the dataset of this study was fixed at 500 × 375 pixels, the width and height of anchor boxes were directly clustered without normalization. The anchor box clustering results of the dataset in this paper are shown in Figure 6. The three-colored point groups in Figure 6 indicate that the initial anchor box values of YOLOv5s are divided into three categories: the largest feature map anchor box, the middle feature map anchor box, and the smallest feature map anchor box. 9 “x” represent the anchor box scale after clustering.

According to the clustering results, the original 9 anchor box scales were replaced with the following scales: [25, 31], [63, 75], [101, 129], [79, 188], [197, 204], [252, 300], [106, 277], [323, 393] and [142, 379].

Generally, cluster centers are assigned to the samples by calculating the Euclidean distance of vectors. However, there is an imbalance error when Euclidean distance is used because the scale of anchor box varies. For a large-scale anchor box, its clustering error is large, and so on [27]. Therefore, similarity was measured by the intersection over union (IoU) between the anchor box and the boundary box. Figure 7 shows two anchor boxes of different scales; the IoU was calculated by aligning the upper-left corner of the boxes, as shown in the formula below.
(1)$$IoU\;=\frac{\min\left(W_{a},W_{b}\right)*\min\left(H_{a},H_{b}\right)}{W_{a}H_{a}+W_{b}H_{b}-\min\left(W_{a},W_{b}\right)*\min\left(H_{a},H_{b}\right)}$$
where *IoU* is the calculated intersection over union; (*W_a_*, *H_a_*) and (*W_b_*, *H_b_*) are the scales of two anchor boxes, respectively. Theoretically, the more similar the anchor box and the boundary box are, the smaller the metric value should be.

#### 3.3.2. CIoU Loss

In the process of calculating the CIoU, we need to understand the relationship between the detection box and the target box, which contains the intersection and union of the detection box and the target box. The detection box, the target box and the relation between them are visually introduced in Figure 8.

Usually, GIoU Loss is used as the bounding box loss function in the original YOLOv5. GIoU Loss is proposed to solve the problem that IoU cannot measure the distance between two boundary boxes and cannot reflect the intersection mode of two boundary boxes. GIoU Loss can better reflect the intersection of boundary boxes by introducing a penalty term, and can effectively alleviate the problem of gradient disappearance in the case of non-overlapping IoU. When there is no overlap between the detection box and the real box, the loss of IoU is the same, and GIoU is added to the C detection frame (C detection box is the smallest rectangular box that contains the detection box and the real box), so as to solve the problem that the detection box and the real box do not overlap. The calculation method of GIoU Loss is as follows [28]:(2)$$GIoU=IoU-\frac{\left|C-\left(B\cup B_{gt}\right)\right|}{C}$$
(3)$$Loss_{GIoU}=1-IoU+\frac{\left|C-\left(B\cup B_{gt}\right)\right|}{C}$$
where *B* and *B_gt_* are the predicted bounding box and the ground-truth bounding box; *C* is the smallest bounding box *C* covering *B* and *B_gt_*.

However, when the predicted boundary box is within the truth boundary box, GIoU Loss is completely consistent and the distance between the two boundary boxes cannot be measured. Therefore, this study adopted a better measurement method of CIoU Loss, including the three geometric measures of overlap area, center distance and aspect ratio, with faster convergence speed and performance [29]. CIoU Loss is shown in the formula below.
(4)$$Loss_{CIoU}=1-IoU+\frac{d^{2}\left(b,b_{gt}\right)}{c^{2}}+\alpha \nu \;$$
(5)$$\alpha =\frac{\nu }{\left(1-IoU\right)+\nu }$$

(6)$$\nu =\frac{4}{\pi^{2}}{\left(\arctan\frac{w_{gt}}{h_{gt}}-\arctan\frac{w}{h}\right)}^{2}$$
where *b* and *b_gt_* are the central points of the predicted bounding box and the ground-truth bounding box, respectively; *d* is the Euclidean distance between the center point of the predicted bounding box and the center point of the ground-truth bounding box; *c* is the diagonal length of the smallest enclosing box covering the two boxes; *α* is a positive tradeoff parameter; *ν* is the consistency parameter of aspect ratio; *w_gt_* and *h_gt_* are the width and height of the ground-truth boundary box, respectively; and *w* and *h* are the width and height of the predicted boundary box, respectively.

### 3.4. Model Performance Evaluation Indicators

In this study, the following indicators were used to quantitatively evaluate the performance of the model: precision (*P*), recall (*R*), mean average precision (*mAP*) and the inference time of a single image. The indicators *P* and *R* are calculated as follows:(7)$$P=\frac{TP}{TP+FP}$$
(8)$$R=\frac{TP}{TP+FN}$$
where *TP* is true positive, that is, the number of target obstacles is correctly detected in the image; *FP* is false positive, that is, the background is wrongly identified as the number of target obstacles in the image; *FN* is false negative, that is, the number of target obstacles is missed in the image.

The *mAP* is the average of the average precision *AP* of all classes as follows:(9)$$AP=\int_{0}^{1}P\left(R\right)dR$$
(10)$$mAP=\frac{\sum_{q=1}^{Q}AP\left(q\right)}{Q}$$
where *Q* is the category of obstacles.

Typically, the processing speed of a detection model can be measured by inference time or frame rate (FPS). Frame rate is a definition in the image domain that represents the number of frames transmitted per second, while inference time is the time taken to process an image. In this study, inference time of a single image was used to measure the speed of the detection model.

### 3.5. Training and Inference Process of the Improved YOLOv5s

#### 3.5.1. Training Process of the Improved YOLOv5s

Firstly, the original image was processed by K-Means anchor box scale clustering, and then the processed image was input into the Improved YOLOv5s model. Subsequently, the predicted boundary box information was obtained according to the anchor box offset value in the Improved YOLOv5s, and the loss of the predicted boundary box was calculated by comparing with the true boundary box; thus, a round of training was completed. The resulting predicted boundary box losses were re-input into the Improved YOLOv5s to optimize the Improved model. Finally, the training process was repeated several times until a predetermined number of rounds were reached and the results were output. The training process of the Improved YOLOv5s is shown in Figure 9.

#### 3.5.2. Inference Process of the Improved YOLOv5s

After training, the weight file with the highest accuracy was obtained, that is, the best weight file. The best trained weight file was used in the Improved YOLOv5s model; at the same time, the image was input into the Improved model. Then, the predicted boundary box information was obtained according to the anchor box offset value in the Improved model. Since there were many generated boundary boxes, it was necessary to use non-maximum suppression (NMS) to eliminate redundant boxes and obtain the best boundary box. Finally, after calculation, detection results were output. The inference process based on the trained Improved YOLOv5s model is shown in Figure 10.

## 4. Results and Discussion

### 4.1. Experimental Configuration and Training

The experimental configuration for this study is shown in Table 2.

The improved model based on YOLOv5-5.0 was initialized with weights trained by the COCO dataset. The *depth_multiple* of the model was set to 0.33 and *width_multiple* to 0.5. In addition, the hyperparameters of the model were set with a batch size of 16, momentum attenuation of 0.9, weight attenuation of 0.0005 and training rounds of 300. For different layers (weight layer, batch normalization layer and bias layer), different learning rate adjustment strategies were used to achieve fast convergence of the model. The initial learning rate was 0.001, and during the warmup stage, one-dimensional linear interpolation was used to update the learning rate of each iteration. After the warmup stage, a cosine annealing algorithm was used to update the learning rate.

### 4.2. Model Performance Evaluation

During the training stage, it can be known whether the model converges through loss changes. After the training, the performance of the model can be preliminarily known by checking the changes of evaluation indexes with the training time, as shown in Figure 11.

Figure 10a shows the change of *mAP* value when IoU was 0.5, and the *mAP* value exceeded 0.84. Figure 10b shows the change in *mAP* value when IoU changed from 0.5 to 0.95 with step size 0.05, and the final *mAP* value reached 0.6. Figure 10c shows the change in precision value, and the final precision value reached 0.75. Figure 10d shows the change in recall rate. The recall rate increased rapidly in the first 50 rounds of training, but after 50 rounds of training, it showed a downward trend accompanied by large fluctuations, and the final recall rate was about 0.82.

### 4.3. Ablation Study on K-Means Clustering Algorithm and CIoU Loss

In order to verify the impact of the K-Means clustering algorithm and CIoU Loss on real-time performance and accuracy, respectively, we conducted four groups of experiments by using the original YOLOv5s algorithm, the original YOLOv5s algorithm only with K-Means, the original YOLOv5s algorithm only with CIoU Loss and the improved YOLOv5s, respectively. The experimental results are shown in Table 3.

In Table 3, after adding the K-Means clustering algorithm to the original YOLOv5s, the inference time of a single image was shortened from 0.062 s to 0.060 s, but the *mAP* value was basically not improved. After replacing the *GIoU* Loss function in the original YOLOv5s with the *CIoU* Loss function, the missed detection and false detection rates were greatly reduced compared with the previous two algorithms, but the inference time of a single image did not have an obvious change. The improved algorithm has faster inference speed and higher detection accuracy. Therefore, it can be seen from the ablation experiment that the K-Means clustering algorithm mainly improves training and inference speed, but only has a limited effect on detection accuracy; the *CIoU* Loss function mainly improves detection accuracy, but has little effect on detection speed.

### 4.4. Comparison between This Study and Other Target Detection Algorithms

The improved YOLOv5s detection algorithm was compared with the Faster R-CNN detection algorithm and the original YOLOv5s detection algorithm under the same training and testing conditions. The comparison results are shown in Table 4 and Figure 12 and Figure 13.

In Table 4, the improved YOLOv5s has obvious advantages compared with the Faster R-CNN detection algorithm and the original YOLOv5s detection algorithm. The inference time of a single image for the improved YOLOv5s was 0.074 s, about 75% lower compared with that of the Faster R-CNN, but the inference time of a single image for the improved YOLOv5s was only slightly increased compared with that of the original YOLOv5s. Compared with the original YOLOv5s, the *mAP* value for the improved YOLOv5s was greatly improved by 5.80%. However, the *mAP* value for the improved YOLOv5s was not significantly different from that of the Faster R-CNN, which was 1.64% lower.

Figure 12 shows the comparison of inference effects between the improved YOLOv5s and the Fast R-CNN. In Figure 11, the *mAP* value of obstacles identified by the Faster R-CNN was higher than that of the improved YOLOv5s. However, it is obvious that some small target obstacles were missed in detection, for example, some trees and haystacks in Figure 12b are not identified.

Figure 13 shows the comparison of detection effects between the improved YOLOv5s and the original YOLOv5s. In Figure 12, the original YOLOv5s with the *GIoU* Loss had the situation of missing and mis-detection during inference process, whereas the improved YOLOv5s with the *CIoU* Loss had the situation of missing and mis-detection greatly reduced. This is because the *CIoU* Loss combines the three geometric measures of overlap area, center distance and aspect ratio, compared with IoU Loss and *GIoU* Loss. For large target detection, using aspect ratio can effectively improve performance, but it does not help to improve the performance of small target detection. Using center distance can effectively improve the performance of small target detection. Therefore, the *CIoU* combined with the three geometric measures has a significant effect on reducing the missing and false detection of the original YOLOv5s, which is also the reason why the *mAP* value of the improved YOLOv5s was improved greatly, compared with the original YOLOv5s.

### 4.5. Comparison between This Study and Other Target Detection Algorithms

Since the improved algorithm is suitable for detecting obstacles in farmland, it was compared with other improved YOLOv5s algorithms which have been used in agriculture in related literature. The comparison results are shown in Table 5.

In Table 5, the three algorithms show advantages in improving the inference speed of a single image compared with Fast-RCNN (see Table 4), although the improved YOLOv5s algorithm in this study is not different from the other two YOLOv5s algorithms that were proposed to improve the inference speed in the related literature. Meanwhile, considering the vibration, strong light and bad weather encountered in the operation of agricultural machinery, the *mAP* of the three algorithms is not high enough for the same multi-type farmland obstacles. However, the improved YOLOv5s algorithm with K-Means and CIoU has obvious advantages in the *mAP* for the same test set, compared with the improved YOLOv5s with SE module and the improved YOLOv5s with Specter module. Of course, none of the three YOLOv5s algorithms are better than Fast-RCNN (see Table 4) in detection precision. In addition, it should be noted that the *mAP* values of the improved YOLOv5s with SE module and the improved YOLOv5s with Specter module are not as good as the results obtained in the related literature, because these results were for single obstacle detection, whereas the dataset in this study is for multi-type farmland obstacle detection.

## 5. Conclusions

To improve the real-time detection performance of multi-type farmland obstacles and reduce the missed detection rate and false detection rate of multi-type farmland obstacles, an improved YOLOv5s algorithm was proposed in this study. The K-Means clustering algorithm was used in the improved algorithm to obtain the initial anchor box value, and thus speed up the convergence speed of model training. Meanwhile, the *CIoU* Loss function, instead of *GIoU* Loss function, was adopted to improve the detection performance of large obstacle targets, while the missing detection rate and the false detection rate were reduced for small obstacle targets. The inference time of a single image for the improved YOLOv5s is three-quarters less than that of the Faster R-CNN, and the *CIoU* Loss combined with three geometric measures of overlap area, center distance and aspect ratio can significantly reduce missed detection and false detection compared with the original YOLOv5s. The *mAP* value of the improved YOLOv5s is higher than that of the original YOLOv5s, and the detection of small targets for the improved YOLOv5s is better than the Faster R-CNN. Due to the small weight file of the improved YOLOv5s, it is easy to be applied and popularized on unmanned agricultural machinery. Of course, it is necessary to establish a more adequate farmland obstacle dataset in future research; thus, it would be helpful to train a better farmland obstacle detection model.

## Figures and Tables

**Figure 1 sensors-22-01790-f001:**
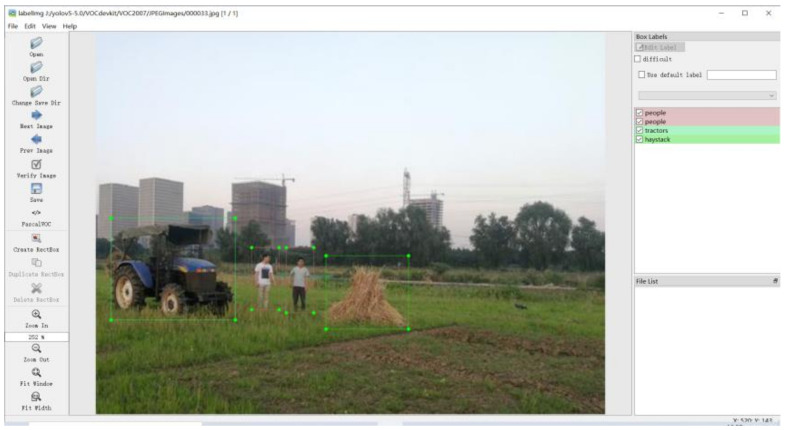
Labeling using LabelImg.

**Figure 2 sensors-22-01790-f002:**
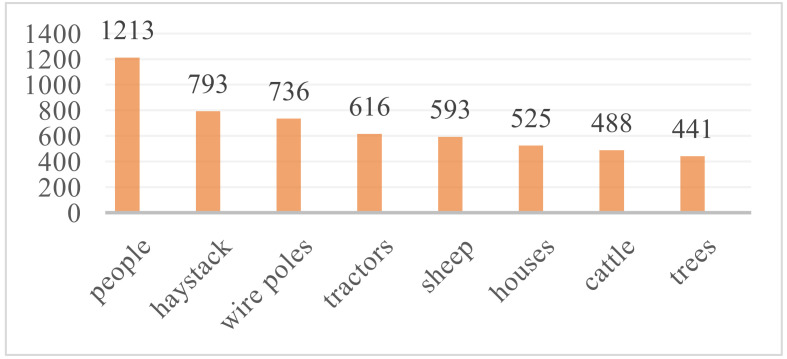
Number of labeling boxes for each type of farmland obstacle.

**Figure 3 sensors-22-01790-f003:**
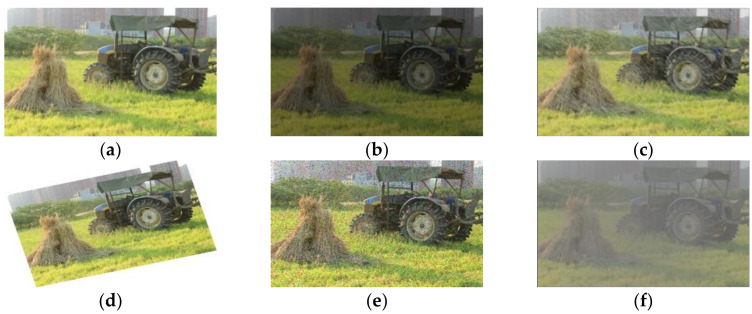
Original image and its expansion processing. (**a**) original image; (**b**) random brightness; (**c**) rainy processing; (**d**) random rotation; (**e**) Gaussian noise; (**f**) atomization processing.

**Figure 4 sensors-22-01790-f004:**
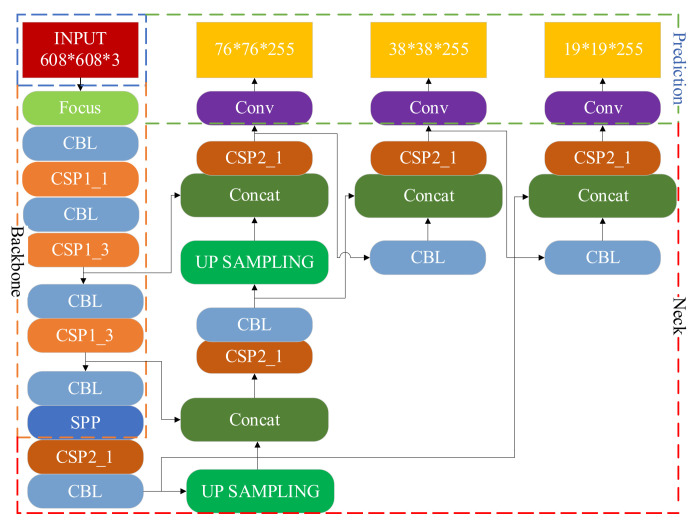
Network structure of the original YOLOv5s.

**Figure 5 sensors-22-01790-f005:**
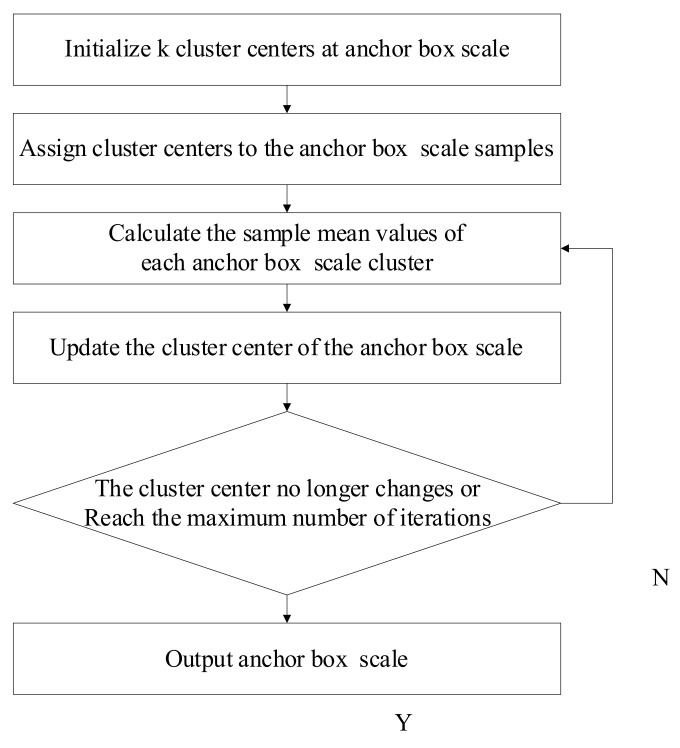
Workflow of the K-Means clustering algorithm.

**Figure 6 sensors-22-01790-f006:**
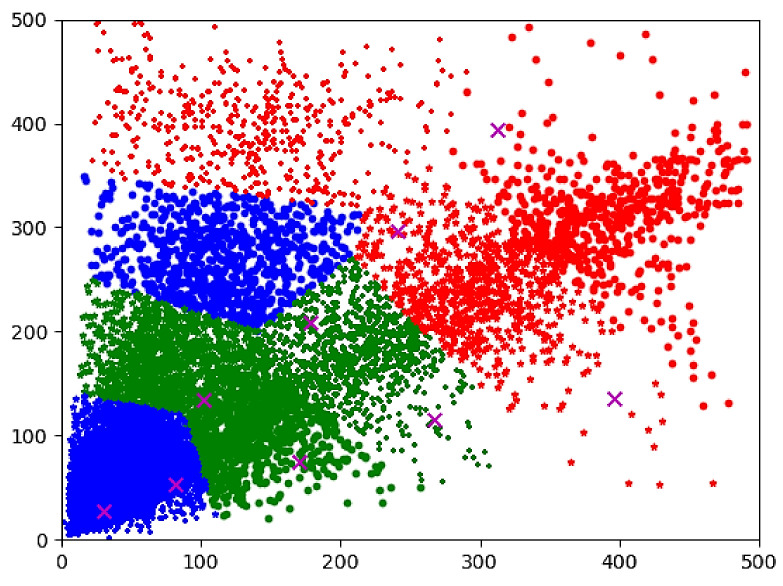
Clustering results of anchor boxes.

**Figure 7 sensors-22-01790-f007:**
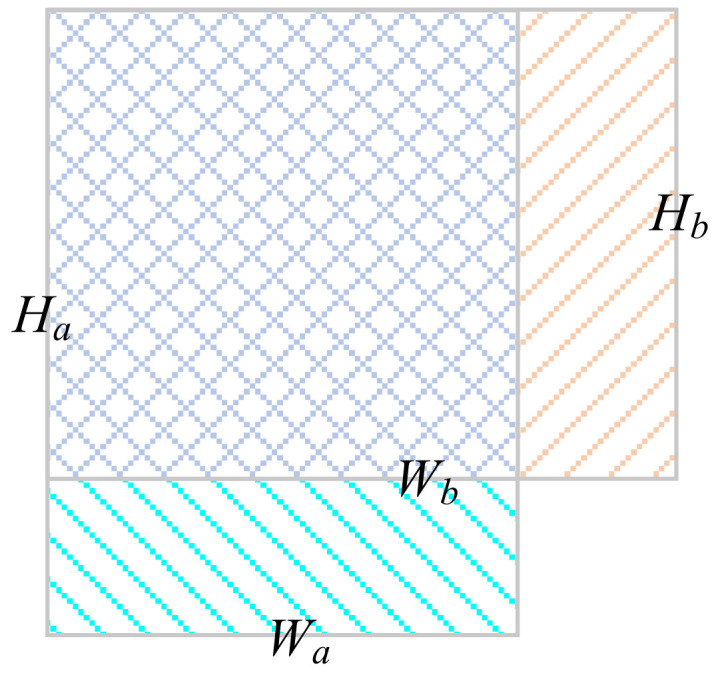
Calculation of IoU.

**Figure 8 sensors-22-01790-f008:**
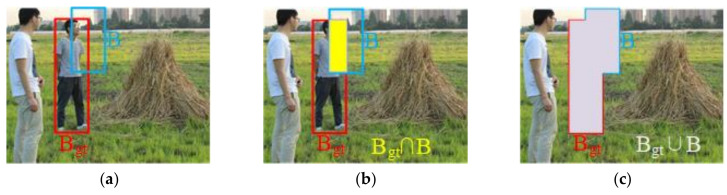
Visualization of the calculation process of CIoU Loss. (**a**) the ground-truth bounding box (B_gt_) and the predicted bounding box (B); (**b**) the intersection of *B* and *B_gt_*; (**c**) the union of *B* and *B_gt_*; (**d**) the smallest bounding box *C* covering *B* and *B_gt_*; (**e**) the union of *C* minus *B* and *B_gt_*; (**f**) c (the diagonal length of *C*) and d (the distance between the center points of *B* and *B_gt_*).

**Figure 9 sensors-22-01790-f009:**
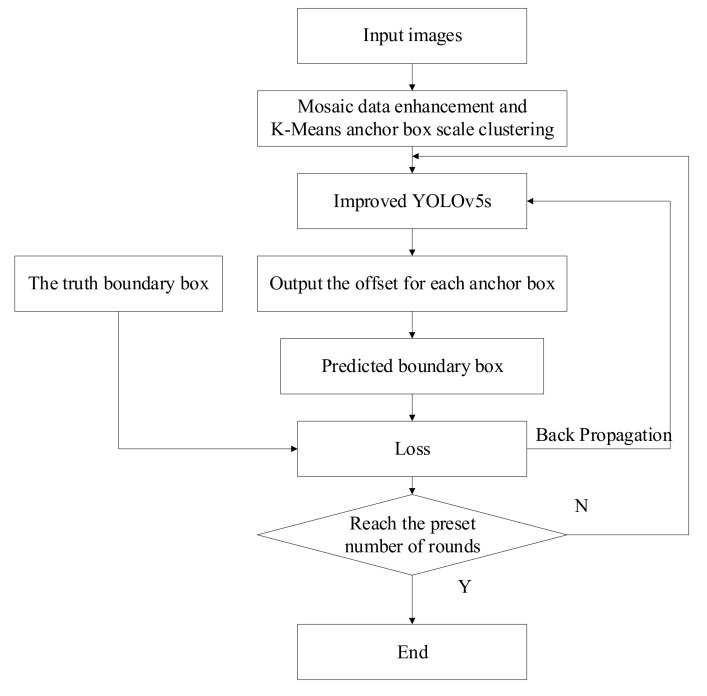
Training process of the Improved YOLOv5s.

**Figure 10 sensors-22-01790-f010:**
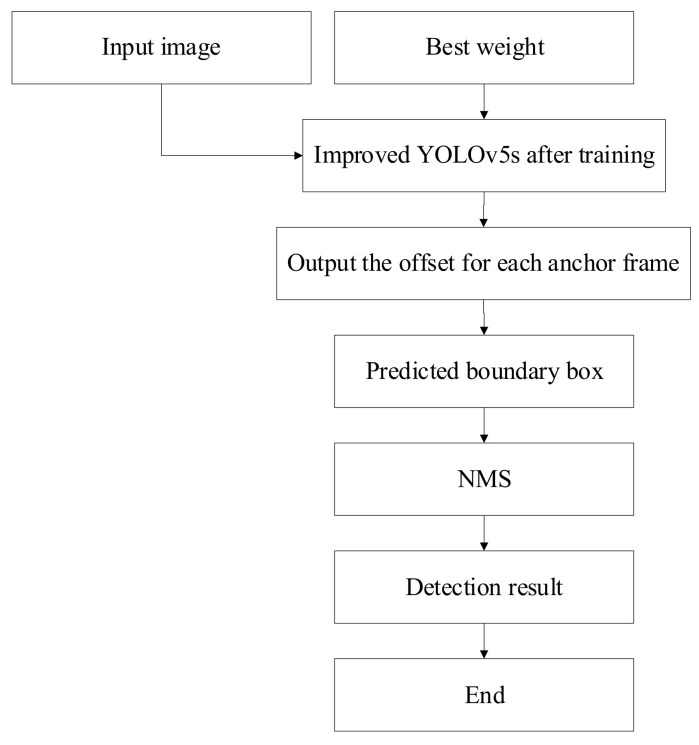
Inference process of the Improved YOLOv5s.

**Figure 11 sensors-22-01790-f011:**
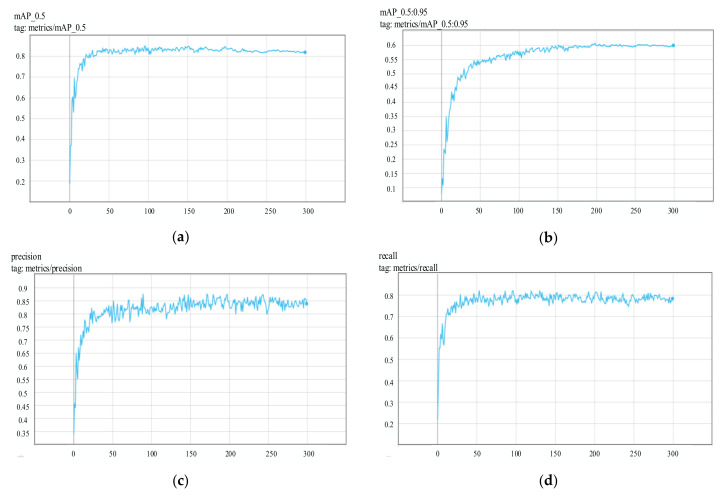
Changes of evaluation indicators. (**a**) *mAP*@0.5; (**b**) *mAP*@0.5:0.95; (**c**) precision; (**d**) recall.

**Figure 12 sensors-22-01790-f012:**
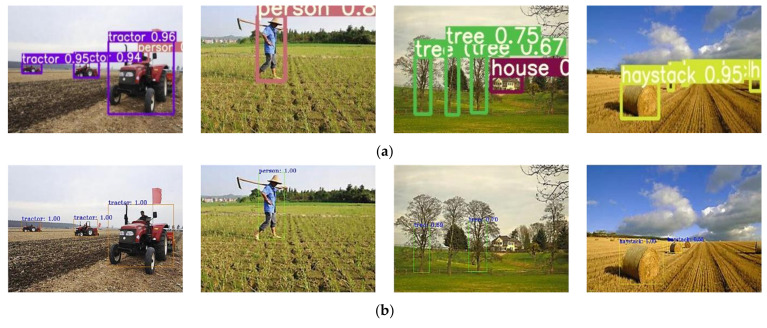
Comparison of detection effects of improved YOLOv5s and Faster R-CNN. (**a**) detection effects of the improved YOLOv5s; (**b**) detection effects of the Faster R-CNN.

**Figure 13 sensors-22-01790-f013:**
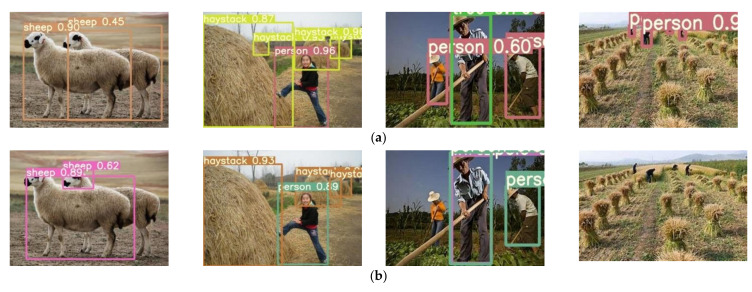
Comparison of detection effects of improved and original YOLOv5s. (**a**) detection effects of the improved YOLOv5s; (**b**) detection effects of the original YOLOv5s.

**Table 1 sensors-22-01790-t001:** Comparison of YOLOv5s models.

Detection Algorithms	Application	Improvement Method
Algorithms in the literature	Main research on single-type target detection and recognition	Adding modules which can make the model of YOLOv5s lightweight
Algorithm in this paper	Recognition of multi-type obstacles in farmland for un-manned agricultural machinery	Using K-Means clustering to speed up convergence and CIoU Loss function to reduce missed detection and false detection

**Table 2 sensors-22-01790-t002:** Hardware and software.

Configuration	Parameter
Operating system	Ubuntu16.04 LTS (Canonical, London, UK)
Graphics card	GeForce GTX1060 6G (NVIDIA, Santa Clara, CA, USA)
CPU	Intel(R) Core (TM) I7-8700K CPU @3.70GHz (Intel, Santa Clara, CA, USA)
Deep learning framework	Pytorch1.6.0
Programming environment	Python 3.6, CUDA9.0, CUDNN7.4.2

**Table 3 sensors-22-01790-t003:** Comparison of effects of K-Means clustering algorithm and CIoU.

Improvement Measures	Inference Time of Single Image (s)	*mAP* (%)
Original YOLOv5s	0.062	59.32
Original YOLOv5s (With K-Means)	0.060	59.61
Original YOLOv5s (With CIoU Loss)	0.071	65.08
Improved YOLOv5s	0.074	65.12

**Table 4 sensors-22-01790-t004:** Comparison of Faster R-CNN, original YOLOv5s and improved YOLOv5s.

Detection Algorithm	Inference Time of Single Image (s)	*mAP* (%)
Faster R-CNN	0.274	66.76
Original YOLOv5s	0.062	59.32
Improved YOLOv5s	0.074	65.12

**Table 5 sensors-22-01790-t005:** Comparison of our improved YOLOv5s and other YOLOv5s models.

Detection Algorithm	Inference Time of Single Image (s)	*mAP* (%)
Improved YOLOv5s (With K-Means and CIoU)	0.074	65.12
Improved YOLOv5s (With SE module)	0.073	58.14
Improved YOLOv5s (With Specter module)	0.076	57.23

## Data Availability

The raw data needed to reproduce these findings cannot be shared at this time, as these data are also part of further research.

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
