# Peer review of "Detection of Farmland Obstacles Based on an Improved YOLOv5s Algorithm by Using CIoU and Anchor Box Scale Clustering"

_sensors, 2022, doi:10.3390/s22051790_

Round 1

Reviewer 1 Report

The authors propose an algorithm for object detection of farmland obstacles. The algorithm is based on YOLOv5s and introduces two optimizations: a clustering of anchor box scales using the K-means algorithm and a CIoU loss function to improve detection accuracy.

The idea is interesting, and the results seem promising but there are some issues that should be clarified to improve the work.

Bout the training, did you consider the same conditions when training faster R-CNN and the original YOLO for your dataset?

Your solution is one Pareto point together with the other two solutions based on Faster R-CNN and improved YOLO. So, the other algorithms are not necessarily worst or better, unless there are some performance or accuracy constraints.

You consider the optimized YOLOv5s for your custom dataset. Should we consider it specific for this dataset or can we generalize its applicability? Please clarify.

It would be important to understand how each of the two proposed optimizations influences the performance and the accuracy. Could you please provide results using each of the optimizations independently?

In section 2.4 you mention other metrics, like P and R. However, there are no results for these metrics. You should include them.

Author Response

Dear reviewer 1

Thanks for Reviewer’ comments concerning our manuscript entitled “Detection of farmland obstacles based on improved YOLOv5s algorithm by using CIoU and anchor box scale clustering” (sensors-1596841).Please see the attachment.

Sincerely yours!

Reviewer 2 Report

This paper proposes an improved YOLOv5s algorithm based on K-means clustering algorithm and CIOU loss function to balance the detection accuracy and detection speed of farmland obstacle detection. This is an interesting research paper. There are some suggestions for revision.

  1. The motivation is not clear. Please specify the importance of the proposed solution.
  2. Please highlight the innovations of the proposed solution in introduction.
  3. Introduction misses some relevant references. "Small Object Detection Method Based on Adaptive Spatial Parallel Convolution and Fast Multi-Scale Fusion", Remote Sensing 14 (2), 420, 2022 and "Convolutional neural network based detection and judgement of environmental obstacle in vehicle operation", CAAI Transactions on Intelligence Technology 4 (2), 80-91, 2019.
  4. The article lacks an overview of the related work research in the fields of farmland obstacle detection and object detection.
  5. The clustering algorithm for anchor boxes described in the subsection "K-means clustering algorithm" is similar to the paper "Redmon J , Farhadi A . YOLO9000: Better, Faster, Stronger. IEEE, 2017:6517-6525.", but the author does not cite this paper.
  6. This is no need to show the network structure of the original YOLOv5s. Please shorten the descriptions of original YOLOv5s network architecture and specify the contributions of the proposed solution.
  7. The subsection "Training process of the improved YOLOv5s" describes the yolov5 training process improperly. The loss used to train the network should not only be CIoU Loss, but also classification loss and regression loss.
  8. What is the definition of beat weight file? Highest accuracy or smallest loss? It is recommended to explain clearly.
  9. More technical details of the proposed solution and the related mathematical analysis and equations should be given.
  10. At present, the open-source YOLOV5 has been updated to six versions. Which version is the method compared in this article?
  11. Faster RCNN has been released for many years, and it is recommended to add target detection algorithms proposed in recent years for comparison, such as YOLOX and NanoDet-Plus.
  12. “Real time” is mentioned in the beginning of this paper. But this paper does not show the real-time performance of the proposed solution.

Author Response

(The authors gave the same response as above.)

Reviewer 3 Report

This paper proposes an improved YOLOv5 object detection algorithm based on the K-means clustering algorithm and the CIoU loss function was proposed to improve the detection precision and speed up the real-time detection.

The following significant details of the proposed approach need further clarification.
Figure 1: It would be good to include farmland obstacles in this figure for the purpose of demonstration.
Figure 3.c: How to perform rainy pre-processing?
Figure 3.d: Is this rotation degree too large? How about the black boundary?
Figure 5: How to determine the number of clusters? There are no experiments to discuss this.
Table 2: Since the proposed approach adds two modules (i.e., CIoU loss and clustering) to the conventional YOLO approach, an ablation study needs to be conducted to verify each module.
There is no discussion on the inference speed. In addition, it is not clear to me how CIoU loss and clustering, which are used in the model training, can help to accelerate the inference speed.

Author Response

(The authors gave the same response as above.)

Round 2

Reviewer 1 Report

The authors have addressed all my issues.

Author Response

Dear reviewer

Thank you again for suggesting revisions.We have checked and corrected some grammatical errors in the paper.

Sincerely yours,

Jinlin Xue

Reviewer 2 Report

All my concerns have been addressed. I recommed this paper for publication.

Author Response

Dear reviewer

Thank you for suggesting revisions again. 

Sincerely yours,

Jinlin Xue

Reviewer 3 Report

Regarding Q5, thanks for adding the experiment on the inference speed in Table 5. However, these inference speeds are very similar to each other. It is not clear to me how CIoU loss and clustering, which are used in the model training, can help to 'accelerate' the inference speed. Please add some discussions to explain this. Otherwise, please remove such a statement from the paper.

Author Response

Dear reviewer

Thank you again for suggesting revisions. Please see the attachment which includes our responses.

Sincerely yours,

Jinlin Xue
